# Indigenous and Ecofeminist Reclamation and Renewal: The Ghost Dance in Silko's *Gardens in the Dunes*

Elizabeth McNeil

Languages and Cultures, College of Integrative Sciences and Arts, Arizona State University,
Phoenix, AZ 85004, USA; mcneil@asu.edu

**Abstract:** Early in the development of ecofeminist literary criticism, white feminists borrowed shallowly and unethically from Indigenous cultures. Using that underinformed discourse to interpret Native American women's literature resulted in idealizing and silencing Indigenous women's voices and concerns. Native American feminist literary critics have also asserted that a well-informed, inclusive "tribal-feminism" or Indigenous-feminist critical approach can be appropriate and productive, in that it focuses on unique and shared imbalances created by white patriarchal colonization, thinking, and ways of being that affect Indigenous and non-Indigenous women and cultures and the environment. In her third novel, *Gardens in the Dunes*, Leslie Marmon Silko interweaves an ecological critique of white imperialist botanical exploitation of landscapes and Indigenous peoples globally with both a celebration of Native American relationships to the land and Indigenous women's resourceful resistance and an ecofeminist reclamation of European pagan/Great Goddess iconography, sacred landscapes, and white feminist autonomy. Expanding on earlier Indigenous-feminist readings, this ecofeminist analysis looks at a key trope in *Gardens*, the Ghost Dance, an environmentally and ancestrally focused nineteenth-century sacred resistance and reclamation rite. Silko's is a late-twentieth-century literary adaptation/enactment in what is the continuing r/evolution of the Ghost Dance, a dynamic figure in Native American literature and culture.

**Keywords:** Leslie Marmon Silko; *Gardens in the Dunes*; ecofeminism; Indigenous feminism; Ghost Dance

> [W]omen who defend the people, women who stand for the Earth . . . understand . . . our lives are intertwined . . . (LaDuke 2004, p. xiii)
>
> No one may be turned away from the gathering, Sister said; otherwise, the Messiah will not come. (Silko 1999, p. 463)

## 1. Introduction: Ecofeminist and Indigenous Experience, Activism, Theory

Before the term was actually coined by d'Eaubonne (1974) in *Le féminisme ou la mort* (Feminism or Death), "ecofeminism" was already associated with Western women's reclamation of their connection to the Earth and the primordial (pre-Christian) Great Mother Goddess as an integral part of an overall reclamation of female pre-history and history; nature-focused spiritual autonomy; physicality, sexuality, reproduction, and health; epistemologies; and voice during the 1960s–1970s social revolution.[1] "Environmental racism" protest activism in the latter half of the twentieth century and initial decades of the twenty-first has been increasingly committed to addressing the toxic living conditions forced on Indigenous people and poor communities in the US and worldwide, and the largely Anglo field of environmental justice literary criticism has been building through its interaction with Native American and other ethnic American literatures. However, the relationship between ecofeminist criticism and Indigenous literature is still a tenuous one.

In light of hundreds of years of white colonization and the fight they must wage against cultural genocide, Native American women may dismiss feminism and ecofeminism outright. Critics of early ecofeminism have argued that the movement was overly

idealistic, focusing too much on a mystical connection with nature, including a stereo-typical imitation of Indigenous reverence for the land, while failing to make of primary concern the actual conditions of women, particularly women of color. In "Ecofeminism and Native American Cultures: Pushing the Limits of Cultural Imperialism?" ecofeminist literary critic Gaard (1993) sees white feminists' "cultural cannibalism" borrowing "from Native American and Eastern cultures the pieces that fit into their theory, while ignoring other aspects of those cultures" (p. 296). Besides shallow and unethical cultural co-opting, ecofeminist theorist Sturgeon (1997) asserts in "The Nature of Race: Discourses of Racial Difference in Ecofeminism" that much of the relatively early ecofeminist discourse of the late 1980s-early 1990s silences Indigenous women's voices "even while idealizing them. This process, besides supporting racism, prevents ecofeminists from effectively envisioning solutions to environmental problems" (p. 269).

Additionally, while white feminists who lack a heritage with strong connections to the land find much upon which to build in ecofeminism, many "Native American women . . . have not needed to build ecofeminist theory because their own cultures provide them with an ample understanding of interconnectedness and interdependence of humans and nature" (Gaard 1993, pp. 295–96). In "Kochinnenako in Academe", scholar and activist Allen (1985) (Laguna Pueblo) asserts that Native American women

> perceive [feminism] (correctly) as white-dominated. They (not so correctly) believe it is concerned with issues that have little bearing on their own lives. They are also uncomfortable with it because they have been reared in an anglophobic world, one that views white society with fear and hostility; but because the fear of and bitterness toward whites and their consequent unwillingness to examine the dynamics of white socialization, American Indian women often overlook the central areas of damage done to tribal tradition by white Christian and secular patriarchal dominance. (p. 85)

For Allen (1985), to study and teach Native American literature requires an intertwined, interdependent approach that she states is "best described as tribal-feminism or feminist-tribalism" (p. 84). Feminism holds the shared potential with Indigenous cultural belief systems to read Native American literature and culture from a gynecentric perspective, as antidote to the hundreds of years of interpretation of Indigenous life through the lens of "paternalistic, male-dominant models of consciousness" (p. 84). Although Native American tribal cultural systems carry "disequilibrium" from white patriarchal colonialism, what prevails is the "belief in balance and relationship and the centrality of women as basic to harmonious, evenhanded ordering of human society" (p. 85).

In "A Gynostemic Revolution", Miranda (2007) (Ohlone-Costanoan Esselen Nation and Chumash) asserts that Native American women's dismissal of Second-Wave feminism has also to do with some white women's devaluing motherhood and rejecting men as part of their resistance to crippling patriarchal power structures. Such sex-based disdain is seen as a sacrilegious imbalancing of the sexes in Indigenous cultures, a fragmentation and weakening of the sacred whole. White women have been caught in the rather untenable position of rejecting the essentializing Western patriarchal notion of all women as closer to nature and therefore subordinate to men and non-nature/technology, while simultaneously attempting to assert female natural, physical, and reproductive "knowing" subjectivity against the deeply entrenched hierarchical, misogynist, racist, and mechanistic Western worldview. Also fighting this life-denying Western mindset, and while enjoying far fewer of its protections and comforts, Indigenous women, on the other hand, assert a fundamental relationship to the Earth for women and men, with the scope of that relatedness generally broader and deeper in works by Indigenous writers. A distinct focus of Native American literature is the emphatic rejection of the relegation of women, indigeneity, and nature to the status of passive victims of toxic "progress", and the concomitant assertion of their subjectivity.

Through "allyship", ecofeminist activism has seen increasing efficacy in the last two decades by becoming more inclusive of Indigenous women's perspectives (see, for exam-

ple, Sempértegui 2021). However, profound differences persist in sociopolitical realities, problems of cultural co-opting by white women, and an historical lack of effective coalition-building between white and Indigenous women. Despite daunting dissimilarities, many Native women and ecofeminists can agree, nonetheless, that, as early ecofeminist theorist and author Griffin (1997) puts it in "Ecofeminism and Meaning", "[t]he racist mind, the misogynist mind, the mind afraid of nature and which denies natural limitation and mortality are often the same mind", an epistemological framework that operates under "the illusion that we who speak and write are not part of nature, not part of each other" (p. 225). Both ecofeminist and Indigenous writers and activists work against this mindset and "have many values in common", contends Gaard (1993), the most prevalent and encompassing of these being the fundamental principle of "the interconnectedness of all life" (pp. 309, 308).

Lucas (2004), in "No Remedy for the Inuit: Accountability for Environmental Harms under U.S. and International Law", notes that ecofeminist theory seeks to "deconstruct racist essentialism" (p. 200). Gaard and Murphy (1998) observe, in their introduction to *Ecofeminist Literary Criticism: Theory, Interpretation, Pedagogy,* that Third-Wave feminism's ecofeminist practices have, indeed, been increasingly rooted in the struggles of poor, colonized women of color worldwide "to sustain themselves, their families, and their communities" in the face of "the 'maldevelopment' and environmental degradation caused by patriarchal societies, multinational corporations, and global capitalism" (p. 2). Critical to growing theory and practice, ecofeminists South and North are working for "environmental balance, heterarchical and matrifocal societies, the continuance of indigenous cultures, and economic values and programs based on subsistence and sustainability" (p. 2). At the same time ecofeminism has grown globally in its pragmatic localized applications, US writers of all ethnic backgrounds have been producing a new post?colonialist creative literature. These works illuminate environmental and human rights abuses, interweaving processes and actions to combat this ongoing degradation through character growth and healing, individual and community relationships, and cultural beliefs and actions—theoria and praxis.

## 2. Reading *Gardens in the Dunes* as Indigenous and Ecofeminist Text

In *American Indian Literature, Environmental Justice, and Ecocriticism: The Middle Place,* critic and environmental justice activist Adamson (2001) notes that our current "environmental crisis involves a crisis of the imagination", and so "writers, teachers, environmentalists, and literary and cultural critics have a key role to play in these conversations and debates" (p. 184). Throughout their careers, Native American women novelists have forwarded the ongoing conversation about our collective responsibility to all of life on Earth. In her third novel, *Gardens in the Dunes,* Silko (1999) interweaves an ecological critique of the nineteenth-century white imperialist botanical exploitation of landscapes and Indigenous peoples with both a celebration of Indigenous relationships to the land and women's resourceful resistance and an ecofeminist reclamation of European pagan/Great Goddess iconography, sacred landscapes, and white feminist autonomy.

Set in the 1890s, mainly in the Southwest, but also significantly on the east coast and in Europe and Brazil, the novel highlights the lives of the self-possessed sisters Indigo and Sister Salt, who are among the last of the fictitious Sand Lizard people, and Hattie, a white feminist religious studies scholar. Hattie has received higher education at a time when women were rarely afforded such access. After her male Harvard professors reject her thesis on "the equal status accorded the feminine principle in the Gnostic Christian tradition" (Silko 1999, p. 99), she meets and marries Edward, a gentleman entrepreneur-scientist who collects plant materials worldwide through the illegal and unethical means typical of imperialist botanists. Edward eventually dies as the result of his inability to perceive, much less divorce himself from, the malevolence of his profession.

After Indigo, the youngest sister, escapes from the southern California Sherman Institute (one of the infamous boarding schools for Indigenous children), she ends up traveling to Europe for the summer with newlywed Hattie and Edward. Over the course of

the novel, influenced by Indigo's commonsensical, dynamic relationship to self and nature, Hattie becomes increasingly resolved in her refusal to adhere to negative Western norms as her apprehension of the divine feminine deepens. Indigo also shows the upper-class Easterner, Hattie, how to be a citizen of the world, which is a way of thinking and being that Silko returns to throughout her oeuvre. In an interview, Silko emphasizes that "the indigenous people of the Americas, we're not only Indian nations and sovereign nations and people, but we are citizens of the world" (Arnold 2000, p. 165). Eventually, as a result of the indigenist epiphanic connection Hattie makes to her own "Old European" animistic heritage as she travels with Indigo, "it was as if her old self molted away", and Hattie realizes, "there was no need for anything more . . . " (Silko 1999, p. 377).

Hattie, whose sexuality, and thus, Silko clearly suggests, spirituality, is the stunted type forced on white women of her time and class, finally realizes, "*there are no sins of the flesh, spirit is everything!*" (p. 450). In contrast, for Indigo and Sister Salt, the awareness of their sexuality and sacred spirit remains intact throughout their lives. The most visible affirmation of the sisters' and Hattie's autonomous physical self and ancient continuity of spirit, which white patriarchal culture has sought to take from all three of them, is their participation in the Ghost Dance, first at the beginning and then also at the end of *Gardens in the Dunes*. Though she has a fractured skull from a brutal beating and rape by a white man in town, Hattie joins Indigo and Sister Salt at the camp nearby where the second ritual is being enacted, though she does not participate directly. The ceremony is interrupted by the sudden presence of Hattie's parents, though Hattie fulfills the environmentally redemptive and spiritually renewing spirit of the Ghost Dance by serendipitously burning down half the town (no one is hurt). Hattie soon moves past the rape trauma to create a new life in Europe studying ancient pagan stone art with her feminist aunt, while Indigo, Sister Salt, and Sister's baby son begin to rebuild their ancestral gardens and community in the desert Southwest.

*Gardens* is set in the 1890s, at the time of the second wave of the "Ghost Dance", and includes what is perhaps the most elaborate and personalized performance of this environmentally and ancestrally focused nineteenth-century sacred resistance and reclamation rite in fiction. The Ghost Dance and other Indigenous dance-based movements afforded tribal peoples a dynamic mode of confrontation with centuries of astonishing loss of life and lifeways—profound, generational trauma and grief. According to Mooney ([1896] 1991), a white ethnographer who studied, wrote about, and participated in Ghost Dance ceremonies in the 1890s, the peaceful Ghost Dance religion served as a civilizing "revolution as comes but once in the life of a race", since the "moral code" of the movement's originator, the shaman and prophet Wovoka (Numu/Northern Paiute)—of not doing harm, not excessively mourning, and not fighting/waging war—resulted, among tribal nations who took to the new rite, in reduced self-mutilation and destruction of livestock and property enacted in traditional mourning practices, and in "mutual brotherly love" where there had been "warlike predatory . . . deadly hatred" (pp. 782–83). Nations who adopted the Ghost Dance had their own name for the ritual (Kehoe 2006, p. 9; Mooney [1896] 1991, p. 791) and adapted its practice to their own cultural expression (see Mooney [1896] 1991). Murray (2007) points out that, in *Gardens in the Dunes*, instead of calling the ceremony the Ghost Dance or giving it any label, Silko's (1999) characters refer to the syncretic ritual as "the coming of the Messiah" (p. 123).[2] Silko's is a late-twentieth-century literary adaptation/enactment in what is the continuing r/evolution of the Ghost Dance, a key, dynamic trope in Native American literature and culture.

In "A Gynostemic Revolution", Miranda (2007) argues that "Silko enables us to read [*Gardens*] as a distinctly indigenous feminist text, and simultaneously urges us toward an understanding of feminist practices that are particular to each culture" (p. 133). Miranda (2007) contends that, through the female characters in this novel Silko's "indigenous feminism" comprises an "indigenous erotics [that] works via the creative life force to re-establish a whole, *balanced* [male and female] energy: to remake the ground, both physical and spiritual" that has been lost to the Western "worship of the destructive" (p. 142).

*Gardens*, with its anticolonial, erotic Indigenous and feminist focus, is an effective vehicle for Silko's message that Indigenous women and white feminists both can—and, more importantly, must—work to rebalance the sacred whole, and even work together, to whatever extent, to do so. To rebalance the fragmented sacred whole, which both Indigenous people and ecofeminists note is critical for planetary health on all levels, Miranda (2007) sees Silko re-opening in *Gardens* "an exchange of knowledge between indigenous communities and contemporary Western feminism ... [in which] Silko intends information to travel both ways ... : into Western feminist communities to communicate strengths from indigenous women, and into indigenous communities to communicate ideological developments from twenty-first century Western feminists" (p. 136).

Critics have also noted how *Gardens in the Dunes* continues the combination of "the local and indigenous with a wider comparativist and even universalizing sweep" (Murray 2007, p. 123) that Silko (1991) had explored in her second novel, the darkly comic *Almanac of the Dead*, though, in *Gardens*, her message is more sympathetically—or, in a suitably nineteenth-century gynocentric and spiritually and morally corrective literary sense, sentimentally—rendered. As noted by Ruoff (2007) in "Leslie Marmon Silko's *Gardens in the Dunes*: Contact Zones and Cross Currents" and Murray (2007) in "Old Comparisons, New Syncretisms and *Gardens in the Dunes*", such "strongly hybrid and transnational" and "comparativist and synthesizing" impulses are not, however, appreciated by all critics of Native American literature (Murray 2007, pp. 119–20). Cook-Lynn (Crow Creek Lakota), for example, rejects the "cosmopolitanism" she sees in "currently popular American Indian fiction" like Silko's because such work, for her, has lost the tribally specific meaning that helps to create a strongly nationalist identity and, thus, authority. For Cook-Lynn, going outside the author's Indigenous national/tribal intellectual scope reduces the efficacy of the work. By failing to focus on "the intellect of a people expressed in literary art ... as the fabric holding people together", the fiction then lacks the political authority and meaning that could add to the fight for recognition of tribal sovereignty (qtd. in Ruoff 2007, p. 7).

In addition to seeing herself and the people of Laguna Pueblo with whom she grew up as citizens of the world, Silko, who is of Laguna, Hispanic, and European ancestry, told an interviewer that, in her work, she is responding to both her European "ancestor spirits" and her Indigenous American inheritances (Arnold 2000, p. 165). In addition to the numerous Western tribes and one Eastern tribe in *Gardens in the Dunes*, the Mormon ghost dancers and farmers, and the religiously syncretic Indian Messiah and family, other multiethnic thematic threads that run through the novel include Black Indians (the Mayan-African woman whose image haunts Edward, and Sister Salt's lover Candy and their baby), African deities, Mexican workers, a Yaqui transborder revolutionary who had been raised by European Gypsies in the US, and the oldest and most elemental figure from Chinese mythology, Sun Wu-Kung, the Handsome Monkey King. The shapeshifting Monkey King, a trickster born of a stone ovum (like the Nanabush trickster figure in Anishinaabe/Ojibwe culture who is rearticulated in Louise Erdrich's novels), continues his global diasporic career in the children's books that comfort Indigo throughout her European journey with Hattie and Edward. The literary Monkey King takes the place of Indigo's beloved real monkey who has had to stay behind, whom Edward had named Linneaus, after eighteenth-century taxonomist Carl Linneaus, the great hierarchical categorizer of all life on Earth—and one of the fathers of "scientific racism".

Each of Silko's three female point-of-view characters crosses boundaries of race, class, culture, and geography in *Gardens in the Dunes* to strengthen and build new possibilities directly related to her own particular female indigeneity. Reading *Gardens* from interwoven Native American Indigenous and ecofeminist perspectives thus seems an appropriate, if not necessary, approach to this text. As tribal-feminist critic and theorist Allen (1985) demonstrated throughout her career, "feminist theory, when judiciously applied to the field [of Native American Studies] makes the error [of previous white male interpretative bias] correctable, freeing the data for re-interpretation that is at least congruent with a tribal perceptual mode even while it is not identical to it" (pp. 84–85). With her female

point-of-view characters and resolution in *Gardens*, Silko suggests the efficacy of the cross-pollination, or at least the global mutual awareness and support, of American Indian and European indigenous philosophical and land-based sagacity and ecofeminist re-vision.

## 3. The Ghost Dance

In Silko's *Gardens in the Dunes*, the garden trope provides a rhetorical, transnational space for her critique of environmental and human rights abuses. The garden generally also serves as a space to unite and transform people (including, of course, the reader) through a sharing of ways of thinking and being, while offering a vivid background for the appreciation of female and cultural autonomy, as well as compassionate, mutually supportive female community. In the novel, the exception and key contrast to the creative and sustaining notion of the garden is London's Kew Gardens, a space that serves the British government in its imperialist global exploitation and manipulation of nature. Because of government involvement, Kew administrators "swore Edward to secrecy" before his disastrous Pará River botanical collecting journey (Silko 1999, p. 127). Throughout the narrative, Silko shows the environmental, cultural, and spiritual damage caused by colonial-era specimen collection, agricultural, and water diversion practices. In Native American literature in general, Adamson (2001) points out that "[t]he garden metaphor is not necessarily a romanticization of earlier, simpler times. It is often a powerful symbol of political resistance" (p. 181), as it is in *Gardens in the Dunes*. In addition to the thematic spectacle of imperial botany that runs through the novel and the quiet counterpoint of the life-sustaining desert landscape in which Indigo and Sister Salt's family lives, the narrative is framed by an environmental and spiritual trope of fundamental significance in Native American literature and culture, the peaceful, intensely political Ghost Dance ritual that promised to return North American lands to pre-Contact abundance (returning animals, the human dead, and the health of land and water) and traditional ways of living, sans the presence and worldview of the destroying whites.

The ceremony scholars refer to as the Ghost Dance or Spirit Dance is linked to earlier nineteenth-century "millenarian" or "revitalization" dance movements in the Western US, and more broadly to prophetic resistance movements beginning with the 1680 Popé Pueblo revolt in the southwest and extending as far northeast as the 1799 Handsome Lake Longhouse Religion. All occurred in reaction to white encroachment, with its attendant genocide and disease that produced astonishing loss of life, tremendous environmental impacts that led to widespread hunger and starvation, and cultural genocide intended to eradicate Indigenous lifeways in favor of turning the communally oriented survivors into Christian nuclear family farmers. The Ghost Dance originated in two waves, in approximately 1870 and 1889, and is based on Numu (Northern Paiute) and other Great Basin peoples' ancient Round Dance, "an ecologically correlated ritual" associated with seasonal pine nut and other food gathering (Hittman 1997, p. 93), syncretized with what scholars view as Christian tenets.[3] The people's connection to the Earth is enacted through the style of the ritual, in which women and men join hands and shuffle their feet to stay closely connected to the land, dancing in the circular direction associated with the seasonal sun cycle. Showing the continuing fundamental significance of this seasonal spiritual communication, in spring 2005, Wovoka's granddaughter, Frieda Brown, spoke of her grandfather and gave the blessing before the Round Dance that would ensure a good pine nut gathering season in the fall (Fogarty 2005).

The Ghost Dance or Spirit Dance originated in 1870 with Fish Lake Jack, a *wodziwob* ("white hair"/"gray hair") or medicine man, and reemerged in 1889 with Wovoka/Jack Wilson, a "weather doctor" like his father before him, Tavibo (or Taivo), who had been an adherent and promulgator of Wodziwob's earlier Ghost Dance (see Kehoe 2006, pp. 32–34; Young 2002, pp. 274–75). A rancher told Mooney ([1896] 1991) that, during a solar eclipse on 1 January 1889, Wovoka had a vision while he was ill with a fever, in which God told him to go back and share with his people the message that they should love one another, work hard and not steal or lie, live in peace with the whites, and perform a five-day Round

Dance at regular intervals (p. 773). According to Young (2002) in *Quest for Harmony: Native American Spiritual Traditions*, Wovoka told Euroamericans that Ghost Dancers who "lived moral lives consistent with Christian teaching" would be rewarded by reunion with their ancestors in heaven, while "[t]o Indians he said that the dead would return to the earth and the old way of life would be revived on earth" sans whites, who, along with non-believer Indians, would be erased through natural cataclysm (p. 276). When the used-up Earth was renewed, the people would no longer experience sickness, and everyone's youth would be restored "with each return of spring, and ... they would live forever" (Mooney [1896] 1991, p. 818). In trance states, the dancers would receive visions of their beloved dead and of the environmentally rich old way of life, and the dancing would hasten the Earth's transformative renewal process.

Wovoka adamantly denied any hostility toward whites and asserted that his message was one of universal peace. He called himself a prophet, messiah, or messenger from God, as Jesus had been in his time. Some of his followers, however, proclaimed "he was the Son of God, Jesus Christ returned for the Indians" (Young 2002, p. 277). Ghost dancers believed whites had tortured and killed Christ the first time he had appeared, so Jesus had gone back to heaven; now he was returning for the Indians. In *Gardens in the Dunes*, Wovoka, as the flesh and blood prophet, joins the Ghost Dance enacted at the beginning of the book. The Messiah and his family, who "almost seem to float" as they descend from the dark hills into the camp toward the end of that ceremony, are Native American (Silko 1999, p. 31). In the novel (as in reality), Mormons who "believed they were related to the Indians" answer the call of Wovoka (p. 44). For "tak[ing] an Indian to be the Messiah", whom Joseph Smith had prophesied would appear in human form in 1890, the Mormons who participate in the ritual are persecuted, as are the Indigenous dancers (p. 45).[4]

The Ghost Dance was an ecstatic demonstration of Indigenous belief in their original way of life, a tangible assertion of a much greater intangible collective human being—past, present, and future—and a creative form of cultural resistance. By the late nineteenth century, Native peoples' continuing devastating losses had produced deep despair. Learning of his vision by traveling to visit with Wovoka or speaking to other tribes who had adopted the rite, as well as from their children who attended intertribal boarding schools, numerous leaders brought Wovoka's hopeful message and ritual practice to their people.[5] Various adaptations of the ritual thus manifested incredibly quickly in a west-to-east *anti*-Manifest Destiny directionality. Since their sacred Sun Dance had been decreed illegal in the 1880s, Lakota were attracted to the Ghost Dance. Despite sensationalist, unsympathetic white journalists' depiction of the Ghost Dance as a dangerous "craze", the ritual was clearly not a war dance, since it was performed by men and women together, rather than by warriors alone.[6] Nonetheless, when thousands of Ghost Dancers amassed in the Badlands in 1890, incendiary yellow journalism, poor management of the situation by Indian agents, and the bloodthirsty zeal of the 7th Calvary (Custer's outfit, which had been humiliated by the Lakota fourteen years earlier) ultimately led to white settler and BIA panic, the murder of Hunkpapa Lakota leader Sitting Bull, and the subsequent killing of approximately 300 Miniconjou Lakota at Wounded Knee thirteen days later, on December 29, 1890, many of whom were Ghost Dancers. The people were traveling to join other Lakota who wanted to find a peaceful resolution to white–Lakota tension. The Wounded Knee massacre is often erroneously referred to as the "Battle at Wounded Knee" or the last "Indian War". For white culture, Wounded Knee and the Ghost Dance have largely served as symbols of a pathetic end of Indian resistance and, in fact, the end of American Indian history itself, in spite of the fact that, as Kehoe (2006) explains in *The Ghost Dance: Ethnohistory and Revitalization*, "the Wounded Knee Massacre in 1890 was only tenuously connected to the Ghost Dance religion and had little, if any, effect on it" (p. 74).

The Ghost Dance ceremony was not eradicated: the Lakota and other Indigenous nations continued to hold Ghost Dance ceremonies after 1890, and, though greatly reduced, the songs and dance have been practiced throughout the twentieth and twenty-first centuries to today. In the introduction to his three-year study of the Ghost Dance among about

twenty tribes, Mooney ([1896] 1991) states that "the dance still exists (in 1896) and is developing new features at every performance" (p. 653). Commissioned by the government, Mooney ([1896] 1991) spent months at a time, from December 1890 to early 1894, with the nations who were performing the ritual (including the Lakota), spoke with Wovoka, collected Ghost Dance songs, took photographs, and participated in Arapahoe and Cheyenne Ghost Dance rituals. Lakota Ghost Dance songs were heard during the 1930s; the Ghost Dance was performed during the 1950s by Canadian Dakota and Wind River Shoshone and performed as a Christian-syncretized movement called New Tidings by Lakota descendants in Saskatchewan in the 1960s (Young 2002, pp. 284–85; also see Hultkrantz 1981; Kehoe 2006). The Lakota Ghost Dance was revived by Leonard Crow Dog during the reoccupation of Wounded Knee in 1973 by the American Indian Movement (AIM), and the ritual was held again a year later on his ancestral land on the Rosebud reservation (see Kehoe 2006; Young 2002). In 2008, the Robinson Rancheria Pomo Indians of California website noted that "the Ghost Dance as well as the grass dance are still practiced during these modern times" by the Eastern Pomo of Lake County ("Pomo Ceremonies" 2008). A colleague who is a Sun Dancer told me a few years ago that he knows a Nevada man who is the keeper of the Ghost Dance for his tribe.

## 4. The Ghost Dance in Native American Literature

The Ghost Dance and Wounded Knee are often twinned tropes that appear together as literally the last chapter in many texts on Indian history, asserting an end to the "Indian Wars" and Native American way of life.[7] In contrast to the popular white historical perspective of the late-nineteenth-century extinction of the Ghost Dance and other organized American Indian resistance, the Ghost Dance, Wounded Knee, and Wovoka continue to serve as compelling touchstones for Native people, including their presence in literature, visual art, music, dance, film, history, criticism, and theory.[8] Activist and academician Churchill (1984) (Creek-Cherokee) states that the Ghost Dance "is a continuing tradition", a way of life or "worldview" that "is integral . . . to the outlook of any traditionalist Indian" (p. 162). In her second novel, *Almanac of the Dead*, Silko (1991) directly affirms Churchill's assertion of the centrality of the Ghost Dance as an enduring and sustaining way of thinking and being. At the International Holistic Healers Convention, Wilson Weasel Tail claims to his audience that "'[t]he truth is the Ghost Dance did not end with the murder of . . . Ghost Dance worshipers at Wounded Knee. The Ghost Dance has never ended, it has continued, and the people have never stopped dancing . . . '" (Silko 1991, p. 724).

## 5. Reclaiming the Gardens: The Ghost Dance in *Gardens in the Dunes*

Appearing two decades after Vizenor 's ([1978] 1990) *Bearheart*, which also integrates the Ghost Dance, Silko's (1999) enactment of two Ghost Dance ceremonies, at the beginning and end of *Gardens in the Dunes*, frames the narrative to offer readers a circular sense of wholeness in regard to Indigenous history and living culture.[9] The ceremonies depicted in the novel occur after the Wounded Knee massacre. Hattie recalls that, "six or seven years before, newspapers reported the Indians claimed to have a Messiah, a Christ of their own, for whom they gathered to perform a dance. Hattie followed the reports in the *New York Times*. It ended rather badly; settlers feared Indian uprisings, and in South Dakota the army killed more than a hundred dancers" (Silko 1999, p. 262).[10] In an interview, Silko reported that she based her depictions of the ritual in *Gardens* on a Ghost Dance held in Kingman, Arizona, in 1893 (Arnold 2000, p. 167).

The five-day ritual at the beginning of *Gardens* is barely concluded when soldiers descend upon the dancers. The second ceremony at the end of the text is interrupted before the final night. Both scenes are set in Needles, California. Through Indigo's perspective, Silko takes the reader into the heart of the first ceremony, which she constructs as loving, peaceful, intertribal, and multicultural. Sister Salt reminds everyone that "[n]o one may be turned away from the gathering . . . ; otherwise, the Messiah will not come" (Silko 1999, p. 463). Indigo learns that "most of the visitors were Walapai and Havasupai,

and of course Paiute; but a few traveled great distances from the north and from the east, because they heard the Messiah was coming . . . " (p. 29), and "[s]mall groups of Mormons came because the Mormons had been waiting for the Messiah's return . . . . Mormons began to dance hand in hand with the other dancers . . . . [P]ainted with white clay and wrapped in white robes, the Mormons looked like all the others" (p. 29). In addition to the Mormon dancers included in the ritual at the beginning of the novel (Mormons are also significant elsewhere in the story), Hattie is an integral member of the Ghost Dance community and, ultimately, of the serendipitously extended ritual at the end of the novel.

In the opening Ghost Dance, which is enacted over the course of twelve pages, Indigo reports the promises of the ritual as she has learned them from a Paiute woman her mother meets.[11] "Jesus promised Wovoka that if the Paiutes and all the other Indians danced this dance, then the used-up land would be made whole again . . . ", Indigo is told. "The dance was . . . peaceful . . . , and the Paiutes wished no harm to white people; but Jesus was very angry with white people. As the people danced, great storm clouds would gather over the entire world . . . . [G]reat winds . . . . would dry up all the white people and all the Indians who followed the white man's ways, and they would blow away with the dust" (p. 23). Jesus had talked to the people, Indigo learns, and said that if people danced "they would be able to visit their dear ones and beloved ancestors . . . . They must not quarrel and must treat one another kindly. If they kept dancing, great storms would purify the Earth of her destroyers. The clear running water and the trees and the grassy plains filled will buffalo and elk would return" (p. 23). The dancers might fall to the ground "shaking and twitching", and then go still, the Paiute woman tells Indigo, but they awake from the desired trance happy because they have seen the Earth reborn (p. 24). During the ritual, the participants

> were careful to drag their feet lightly along the ground to keep themselves in touch with Mother Earth . . . moving from right to left because that was the path followed by the sun . . . Wovoka wanted them to dance because dancing moves the dead. Only by dancing could they hope to bring the Messiah, the Christ, who would bring with him all their beloved family members and friends who had moved on to the spirit world after the hunger and the sadness got to be too much for them. (p. 26)

Here, as in her other work, Silko suggests to readers that another kind of knowledge should be valued. The Ghost Dancers' merging with the spirit world can be viewed as what ecofeminist philosopher Spretnak (1997) terms an experience of "radical nonduality", an apprehension of a unitive or holistic state of being of the sort that has "been marginalized and devalued by the [Western] modern, objectivist orientation", even among ecofeminists (p. 429). In the novel, the powerful, intrinsic spiritual and physical relationship to the Earth, to each other, and to all-time is, at least for the period of the dance, renewed. As the ritual ends, the Holy Mother and the Messiah's wife open their shawls and "plump orange squash blossoms tumble to the ground" for the hungry dancers, and the Messiah and the Holy Mother speak to each dancer in her/his own language (Silko 1999, p. 31). A Paiute man explains to Indigo that, "[i]n the presence of the Messiah and the Holy Mother, there was only one language spoken—the language of love—which all people understand, because we are all the children of Mother Earth" (p. 32). Silko is arguing that the unitive experience of the Ghost Dance provides the nourishment of interconnection to the participants.

In *Gardens*' initial ceremony scene, Indigo hears the dance just finishing when soldiers, like "giant insects swarming down the hills", descend upon the camp (Silko 1999, p. 35). This insect metaphor is one of Silko's figurative reversals in the story. Racist and anti-Semitic whites have long used rodent and cockroach metaphors in reference to ethnic Others, a dehumanizing figuration that directly correlates to brutal policies and genocidal treatment of members of those ethnic groups. While the soldiers descend, "Wovoka led the dancers in the final rituals of the dance: they all must clap their hands and shake and wave their shawls vigorously to repel diseases . . . " (p. 32).[12] As she and Sister Salt run, Indigo sees "white men . . . seizing the Mormons . . . [and Indian] dancers running in all

directions with Indian policemen chasing them" (p. 32). Sister Salt sees the Messiah and his family as they calmly cross the fierce river in plain view of, but unseen by, the soldiers. Though their mother is never heard from again (Indigo later realizes she has gone to join the Messiah—died), the girls escape and make their way back to their ancestral gardens in the dunes, where their Grandmother Fleet also eventually returns.

In between the novel's opening and closing Ghost Dance rituals, the Messiah is invoked throughout the story, especially by Indigo, who sees evidence of him and of the Holy Family all along her journey to the eastern US and Europe. The Holy Mother (the Mother of God, who is, thus, the female originating deity) also appears throughout the text, in the syncretic Ghost Dance and in the manifestation of the contemporaneous episode of European Mariolatry in Corsica, as well as the abundant archaeological evidence the characters encounter in Britain and Italy of prehistoric goddess worship. Memories of the Ghost Dance ritual and Messiah become for Indigo what poet Peacock (2001) calls a "portable state of grace", referring to a state of being Peacock has carried within herself from girlhood, a continuous blooming of her "true self" from her memories of the absorbing peace of her grandmother's garden (p. 134).[13] Memories of the Messiah and Ghost Dance ground Indigo as she travels through Europe, providing a spiritual link that she perceives in the landscape, people, and the people's beliefs. In Corsica, for example, after Indigo and Hattie see the glowing miracle of the appearance of the "Blessed Mother" Mary, "Indigo was much heartened" (Silko 1999, p. 320). Even though she had not seen "the Messiah or the rest of the family or her mother with the dancers", as she had hoped she would, "all who are lost will be found, a voice inside her said; the voice came from the Messiah, Indigo was certain" (p. 320).

Indigo's seemingly unshakeable faith and self-confidence related to the Messiah becomes a mechanism that not only helps her enjoy and benefit from the trip; her intact self-awareness also helps Hattie. At the second Ghost Dance ritual, which occurs at the end of the novel, Hattie seeks an explanation for the mysterious light she has seen and dreamt of throughout her journeys with Indigo. Sister Salt explains that "[t]he light [Hattie] saw was the morning star, who came to comfort her" (p. 469). Hattie wants to know how she could have seen "the same light in the garden in England and in a dream on board the ship", so Sister explains that "the Messiah and his family traveled the earth—they might be seen anywhere" (p. 469). Inspired by Indigo's grace, Hattie gradually gains a strong and sustaining sense of self/Self that includes an awareness of herself as both autonomous and as a relational being, as a woman and as a being in nature—knowledge from which she has been disenfranchised throughout her life by prevailing Victorian-era social-scientific hierarchical patriarchal mores. Silko's Indigenist and commonsensical assertion of children's intelligence and their inherent valuing of relationship to self, community, and nature is an appreciation shared by ecofeminists (see, for example, Kurth-Schai 1997).

At the end of *Gardens*, the second Ghost Dance ceremony is enacted over the course of eleven pages. Neither Wovoka nor the Mormons attend this time. Wovoka "could not be there because the soldiers wanted to arrest him" (Silko 1999, p. 464) and "no old-time Mormons showed up like they had last time; but who could blame them after their punishment?" (p. 462). However, the Sand Lizard girls, along with Paiutes, Chemehuevis, Mojaves, and Walapais, are there, and all their languages are again understood by everyone, as had occurred during the earlier ceremony. Hattie, who, observes Sister Salt, "might not recover" from the rape and vicious beating, makes her way to where Indigo and her family are camped at the dance site (p. 463). Though too weak to take part in the Ghost Dance, Hattie is accepted as part of the community enacting and benefitting from the rite. When Hattie's father and mother appear with Indian policemen and white soldiers, Hattie realizes the dance is being broken up before the required fourth night because of her presence: "The dancers' prayers saved her life—each night of the dance she recovered a bit more as the Messiah drew nearer. She wept with fury when she saw her mother and the lawyer

whisper to each other—they believed she was ill, out of her head" (p. 471). And now she is the cause of the dancers not getting to see the Messiah.

Forced by her parents to leave, Hattie slips out of their buggy and down an alley in the town of Needles. Because the last night of dancing has not occurred, the Ghost Dance ceremony is incomplete, though a certain sort of completion is offered by Hattie in the next scene, when she lights the matches she finds in the pocket of the lawyer's overcoat that her father had put around her. Realizing she has stumbled onto the livery stable of the man who had raped and tried to kill her, Hattie (ironically cloaked in the mantel of a representative of white legal *in*justice) burns down the rapist's family stable and half the white settler town of Needles. The townspeople had earlier refused to divulge her attacker's identity. This act of retributive justice is only materially violent; neither horses nor townspeople are injured, the text stresses. By virtue of its lack of physical violence and yet successful disruption of malicious white presence, Hattie's unpremeditated act is in keeping with the scourging justice promised through the Ghost Dance. Standing outside the town with the horses she had freed from the stable, Hattie watches, "amazed and elated by the beauty of the colors of the fire against the twilight sky" (p. 473). The colors remind her of the carved Roman gemstones purchased on their trip to England, which she had treasured, and which the rapist had taken.

The rape is a final manifestation of a lifetime's loss of power for Hattie as a woman in white nineteenth-century US society. In the outdoor scene with the liberated animals, Silko suggests that Hattie, through her own confrontation with death (which is linked to her involvement with the Ghost Dance), has gained access to her innate/natural power. Hattie's newly embodied knowledge and experience suggest that this is a "conversion" narrative in which white female empowerment has occurred through a confrontation with white society's most vicious flaws and with Indigenous practices that demonstrate creative, empowering, persistent resistance to the life-denying propensities of white/Western culture. An ironic aspect of her recovery is that Hattie's transformation (or conversion) from semi-enlightened colonial patriarchy to Old European indigeneity is achieved only after she had been beaten over the head with a piece of the "heavy iron rock" meteor whose collection has made it symbolic of the capitalist scientific materialism that doomed her misguided husband (p. 456). The symbolic conversion had continued when she was found, wandering naked (her elemental and sexual self revealed), by Indian people (not whites) who generously give her a dress and take her to town, though this interrupts their lives, and their kindness to Hattie makes them vulnerable to the racist and misogynist townspeople. The white Needles residents had offered minimal sympathy, denigrated the kindness of the people who saved her by calling what she is wearing a "squaw dress", and protected the identity of her attacker. Afterward, Hattie realizes, "it wasn't terribly different from the way it was done in Boston. Now it was clear to her, she could never return to her former life of lies" (p. 459). Besides the Ghost Dance ceremony's affording Silko's character a transformative opportunity for much fuller engagement with life, the open-endedness in the novel of this second Ghost Dance ritual also creates a space for the reader to invest in its completion and cyclical sense of continuum.

## 6. Conclusion: Indigenous and Ecofeminist Emancipatory Strategies

In *Gardens in the Dunes*, as in Native American literature generally, the Ghost Dance is a focal vehicle through which to offer continuing, creative, dynamic spiritual and political resistance to physical and psychological colonization. Toward the end of *Almanac of the Dead*, Silko (1991) clarifies through her character Wilson Weasel Tail what she believes is the true significance of the Ghost Dance: "'Moody and the other anthropologists alleged the Ghost Dance disappeared because the people became disillusioned when the ghost shirts did not stop bullets and the Europeans did not vanish overnight. But it was the Europeans, not the Native Americans, who had expected results overnight . . . '" (p. 722). As ecofeminists might well point out, and as Silko (1996) notes in *Yellow Woman and a Beauty of the Spirit*, the process of the removal of Western capitalist values and pollution has "already begun to happen, and . . . it is

a spiritual process that no armies will be able to stop" (p. 125). In "Revolutionary Enunciatory Spaces: Ghost Dancing, Transatlantic Travel, and Modernist Arson in *Gardens in the Dunes*", Regier (2005) asserts that the syncretic Ghost Dance, "[a]s part of a larger apocalyptic renewal of a world decayed by rapid industrialization" over the past 200 years, works in Silko's *Gardens*, and in other texts that include the trope, "as a . . . discourse in the service of a pantribal renewal in the Ghost Dancing movement . . . . [and] an ongoing revolution in the present" that "affects human beings who live in the Americas as part of the topography of the Americas, . . . a slow-burning, unstoppable change . . . " (p. 153).

The wider environmental, physical, and spiritual implications of the Ghost Dance trope also help Silko create intimate relationships between Indigenous and white women in *Gardens*, and offer a way through which her female characters can maintain or recover the personal power to reject colonization and patriarchy and live their own authentic lives. As Gaard (1993) discusses, coalition-building in the US between Indigenous women and white ecofeminists is only possible when white women seek to develop real relationships with Indigenous women, and active, accurate knowledge and appreciation of Indigenous women's histories and lives in context (p. 310), as Hattie does to a significant extent in *Gardens in the Dunes*. At least imaginatively, Silko offers this sort of cultural "'world-travelling'" (Gaard 1993).[14] for her characters and readers in the novel. Miranda (2007) concludes that, with *Gardens*, Silko "insists on a . . . revolution in which indigenous feminism works to decolonize such systems as patriarchy and paternalism and encourage symbiotic interchange between equals as the actions that can help us all in our perpetual act of finding, and maintaining, balance" (p. 147). Silko reminds her readers that feminists and ecofeminists must seek out knowledge of their own indigeneity or they will remain spiritually lost and, thus, far less powerful than they could be in fighting patriarchal and capitalist systems to heal and balance relationships between women and men, and humans and the environment.

As do the characters in Silko's *Gardens in the Dunes*, Hogan's (1995) (Chickasaw) teen protagonist in her ecocritical novel *Solar Storms* draws sense and strength from the Ghost Dance. To be healed back to life, the traumatized Angela comes to live with relatives in an Indigenous and mixed-blood community made up mostly of elder women who are wise, brave, strong, and loving. "For my people", Angela muses toward the end of the novel,

> the problem has always been this: that the only possibility of survival has been resistance. Not to strike back has meant certain loss and death. To strike back has also meant loss and death, only with a fighting chance. To fight has meant that we can respect ourselves, we Beautiful People. Now we believed in ourselves once again. The old songs were there, came back to us. Sometimes I think the ghost dancers were right, that we would return, that we are still returning. Even now. (Hogan 1995, p. 325)

Silko's (1999) enactment of the Ghost Dance in *Gardens in the Dunes* is, as Hogan's protagonist suggests, an act of resistance and regeneration.

In "Ecofeminist Literary Criticism", Legler (1997) delineates seven "emancipatory strategies" she notes women writers using whose work is focused on the human relationship to the land—strategies Silko employs in *Gardens in the Dunes*:

1.  "Re-mything" nature as a speaking, "bodied" subject.
2.  Erasing or blurring boundaries between inner . . . and outer . . . landscapes, or the erasing or blurring of self-other . . . distinctions.
3.  Re-eroticizing human relationships with a "bodied" landscape . . . .
4.  Historicizing and politicizing nature . . . .
5.  Expressing an ethic of caring friendship, or "a loving eye", as a principle for relationships with nature.
6.  Attempting to unseat vision, or "mind" knowledge, from a privileged position as a way of knowing, or positing the notion that "bodies" know.
7.  Affirming the value of partial views and perspectives, the importance of "bioregions", and the locatedness of human subjects. (pp. 230–31)

Through "the ethic of caring friendship" embedded in and enacted through the Ghost Dance and through the "partial and local knowledge" of Indigo's child perspective and the very specific settings described in the novel, Silko integrates Indigenous-feminist and ecofeminist "emancipatory strategies" to re-myth, re-embody, historicize, and politicize nature; to blur dualistic boundaries of all kinds through her characters' experiences of "radical nonduality"; to re-eroticize Hattie's and the reader's relationship to the land; and to unseat the primacy of logos in order to appreciate knowledge of the body and nature. In *Gardens*, Silko returns subjectivity to white women and to Indigenous women and girls, to indigeneity as philosophically and pragmatically integral to environmental justice acumen, and to nature as a "speaking, 'bodied' subject".

Miranda (2007) argues that Silko's "response to the paternalism in Western feminism ... offers this idea that we are all indigenous ... " to the Earth (p. 146). For Silko, "being indigenous is not purely a question of blood", continues Miranda (2007),

> but of responsibility to the relationships between earth and human spirituality, and ... while our paths to fulfilling that relationship may be different based on our biological places of emergence, in the end it is of primary importance to us all .... [W]e are one race—the human race .... because ... we are *all* indigenous to this planet .... [C]laiming indigeneity requires acceptance of responsibility, constant attention to balance and intent, in ways most non-Indians have rarely contemplated. (p. 146)

As in Silko's (1991) earlier *Almanac of the Dead*, her paradigmatic Ghost Dancing shift in *Gardens in the Dunes* toward collective responsibility supports the idea of words and stories as necessary for creating a sustainable future for all life in our increasingly obviously inter-connected global ecosystem, in which "[n]o one may be turned away from the gathering" (Silko 1999, p. 463). Creating a dynamic, revitalizing shared literary-ritual space in *Gardens*, in which words "in themselves have the power to make things happen" (Bierhorst 1983, p. 3), Silko invokes in readers the knowledge that we are "part of nature, ... part of each other" (Griffin 1997, p. 225). All we need to do is open our eyes and minds, and act.

**Funding:** This research received no external funding.

**Institutional Review Board Statement:** Not applicable.

**Informed Consent Statement:** Not applicable.

**Data Availability Statement:** Not applicable.

**Conflicts of Interest:** The author declares no conflict of interest.

## Notes

1   For a discussion of use of the term previous to d'Eaubonne, see Gaard (1996).

2   On the Ghost Dance in the arts as dynamic process, rather than as a potentially limited and delimiting trope, see Smith (2014).

3   For a reading of *Gardens in the Dunes* in relation to the current environmental catastrophe of global warming, see Tillett (2020).

4   The plural-marriage old-time Mormon widow Mrs. Van Wagnen befriends the girls' Grandmother Fleet. The new Mormons kill Mrs. Van Wagnen's husband. The text suggests they later destroy her house and orchards, and one must suspect that the "terrible odor" Indigo and Sister Salt detect from the barn is the widow's decaying body (Silko 1999, pp. 44–45). Regarding the Mormon presence and possible cross-cultural influences in the Ghost Dance, see Barney (1986), *Mormons, Indians, and the Ghost Dance Religion of 1890*, as well as Kehoe (2006), Mooney ([1896] 1991), and Young (2002).

5   Porter (2007) notes that "[a]lthough Indian schools are presented negatively in *Gardens*, they were one of [the] key means by which news of the Ghost Dance spread" (p. 69n). Silko suggests that cross-cultural botanical exchanges also could have occurred through boarding school contact (Arnold 2000, pp. 163–64).

6   See Reilly (2003) regarding both hostile and sympathetic news reporting of this period, including coverage for Omaha's *World-Herald* by America's first female Native American war correspondent, Suzette La Flesche or Bright Eyes.

7   See, as one of many examples, Utley and Washburn (1977), *Indian Wars*, which closes with Sitting Bull's body: "While in the agency cemetery an infantry company fired three volleys over the graves of slain Indian policemen and a bugler sounded taps, at

the Fort Yates cemetery a detail of prisoners unceremoniously shoveled dirt into an open grave. In it was a rough wooden box containing the canvas-wrapped remains of Sitting Bull" (p. 301).

8   Wovoka and the Ghost Dance are the subjects of many poems. See, for example, Seale (2000), *Ghost Dance*. In fiction, among the most well-known examples are, chronologically, Vizenor ([1978] 1990), *Bearheart*; Silko (1991), *Almanac of the Dead*; Alexie (1993), *The Lone Ranger and Tonto Fistfight in Heaven*; Power (1994), *The Grass Dancer*; Owen (1994), *Bone Game*; Hogan (1995), *Solar Storms*; Alexie (1996), *Indian Killer*; Silko (1999), *Gardens in the Dunes*; and Erdrich (2003), *The Master Butchers Singing Club*.

9   Her framing device in *Gardens* also suggests an understanding of time that Silko (1996) explains in *Yellow Woman and a Beauty of the Spirit*: "For the old-time people, time was round—like a tortilla" in which the dead existed in a parallel dimension with the living, thus indicating that "[a]ll times go on existing side by side for all eternity. No moment is lost or destroyed . . . . The past and the future are the same because they exist only in the present of our imaginations" (p. 137).

10  Here, Silko (1999) purposefully notes Hattie's apprehension of the event through the white newspaper's downplaying of the massacre. Today, it is commonly believed that at least 250 Lakota died at Wounded Knee, with others dying of their wounds soon after. Up to 500 soldiers surrounded Big Foot's camp of destitute, starving people (two-thirds of them women and children), and of these 39 soldiers were wounded and 25 killed (Utley and Washburn 1977, p. 299), some by "friendly fire", since the Hotchkiss guns were positioned across from each other. The most often cited number of Lakota in Big Foot's camp at Wounded Knee is 120 men and 230 women and children, from the military's tally of the group (see Mooney [1896] 1991, pp. 870–71, for more various reports). Of these, the most often cited number of those who died at Wounded Knee or later of their wounds is 300, with some scholars including in that number those who died with Sitting Bull during his attempted arrest two weeks earlier, as well as those who died in several encounters following Wounded Knee. Scholars commonly note that the white soldiers mercilessly hunted down and slaughtered women and children up to three miles away from the Wounded Knee camp, a mission that took hours after the initial machine-gunning. "The most elaborate cover up" of the massacre was the awarding of 29 Congressional Medals of Honor to soldiers who had participated in "the last Sioux campaign; twenty-three were awarded specifically for action at Wounded Knee!" (Miller 1985, p. 270). Numerous websites call for the rescinding of those medals. Mooney ([1896] 1991) notes that this 1890 military offensive against the recalcitrant Lakota, which resulted in "49 deaths on the government side, including 7 Indians and a negro" and "about 300 or more Indians", cost "$1,200,000, or nearly $40,000 per day, a significant commentary on the bad policy of breaking faith with Indians" (pp. 872, 891, 892). Because only one white non-soldier was killed, "and no depredations [had been] committed off the reservation", Mooney ([1896] 1991) terms the white settlers' panic that helped lead to the massacre at Wounded Knee as "something ludicrous" (p. 892).

11  These promised results are in keeping with those noted in Mooney's and others' studies of the ritual.

12  According to Kehoe (2006), the shaking of shawls and blankets on the morning of the fifth day was "to symbolize driving out evil" (p. 6).

13  The "portable state of grace" offered by the Ghost Dance is compelling for Indigenous writers and for artists in other media, Indigenous and non-Indigenous. See Smith (2014), who, in "'Ghost Dance' and the Crisis of Categorization in Indigenous Art", asserts the continuing dynamic presence and process of the Ghost Dance in Indigenous arts as ongoing enactment of creative resistance.

14  Gaard (1993) is citing concepts from Lugones (1990), "Playfulness, 'World'-Travelling, and Loving Perception".

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
