# Peer review of "Indigenous and Ecofeminist Reclamation and Renewal: The Ghost Dance in Silko’s Gardens in the Dunes"

_humanities, doi:10.3390/h11040079_

Round 1

Reviewer 1 Report

This is a good close reading of Silko’s novel that combines ecofeminist and indigenous studies approaches. 

Author Response

Thank you.

Reviewer 2 Report

This article is exceptionally well written.

Author Response

Thank you.

Reviewer 3 Report

This article frames its analysis of Leslie Marmon Silko's novel Gardens in the Dunes in the perspective of ecofeminist and Indigenous theory, in contrast with the now established field of environmental justice literary criticism. It raises the kind of issues that (white) feminism poses for Indigenous women, such as essentialisation (for instance women's close relation to nature, women as victims), and the need for Indigenous women to respond to this by re-inventing their links with white feminists, a form of cosmopolitanism that some Indigenous critics find weakening to the Indigenous cause. The Ghost Dance plays a central part in the story in bringing together Indigo (of the fictitious Sand Lizard people) and Hattie, a white feminist scholar whom Indigo travels with. The Ghost Dance, which in the novel is referred to as "the coming of the Messiah", is also an act of cultural resistance among other 19th-century revitalisation dance movements in the US. More broadly, the author jointly analyses the Indigenous and ecofeminist strategies of emancipation found in the novel, blurring the boundaries between individuals and revitalizing the view that we are all part of nature.

- l. 17-23: This sentence is too long and should be broken into two.

Corrections needed:

l. 12: La Le féminisme ou la mort

l. 63: delete also

Author Response

Thank you. I will make those corrections.

Reviewer 4 Report

The author makes a compelling case, deftly balancing close reading and a command of the secondary literature. 

Author Response

Thank you.